# Smart Sleep Monitoring: Sparse Sensor-Based Spatiotemporal CNN for Sleep Posture Detection

**DOI:** 10.3390/s24154833

**Published:** 2024-07-25

**Authors:** Dikun Hu, Weidong Gao, Kai Keng Ang, Mengjiao Hu, Gang Chuai, Rong Huang

**Affiliations:** 1School of Information and Communication Engineering, Beijing University of Posts and Telecommunications (BUPT), No. 10 Xitucheng Road, Haidian District, Beijing 100876, China; hdingding@bupt.edu.cn (D.H.); gaoweidong@bupt.edu.cn (W.G.); chuai@bupt.edu.cn (G.C.); 2Institute for Infocomm Research, Agency for Science, Technology and Research (A*STAR), 1 Fusionopolis Way, #21-01 Connexis (South Tower), Singapore 138632, Singapore; hu_mengjiao@i2r.a-star.edu.sg; 3College of Computing and Data Science, Nanyang Technological University, 50 Nanyang Ave., Singapore 639798, Singapore; 4Department of Respiratory and Critical Care Medicine, Peking Union Medical College Hospital, Chinese Academy of Medical Sciences and Peking Union Medical College, No. 1 Shuaifuyuan Wangfujing, Beijing 100730, China; huangrong0212@163.com

**Keywords:** sparse sensor-based, sleep posture detection, model-based feature extraction, spatiotemporal convolutional network

## Abstract

Sleep quality is heavily influenced by sleep posture, with research indicating that a supine posture can worsen obstructive sleep apnea (OSA) while lateral postures promote better sleep. For patients confined to beds, regular changes in posture are crucial to prevent the development of ulcers and bedsores. This study presents a novel sparse sensor-based spatiotemporal convolutional neural network (S^3^CNN) for detecting sleep posture. This S^3^CNN holistically incorporates a pair of spatial convolution neural networks to capture cardiorespiratory activity maps and a pair of temporal convolution neural networks to capture the heart rate and respiratory rate. Sleep data were collected in actual sleep conditions from 22 subjects using a sparse sensor array. The S^3^CNN was then trained to capture the spatial pressure distribution from the cardiorespiratory activity and temporal cardiopulmonary variability from the heart and respiratory data. Its performance was evaluated using three rounds of 10 fold cross-validation on the 8583 data samples collected from the subjects. The results yielded 91.96% recall, 92.65% precision, and 93.02% accuracy, which are comparable to the state-of-the-art methods that use significantly more sensors for marginally enhanced accuracy. Hence, the proposed S^3^CNN shows promise for sleep posture monitoring using sparse sensors, demonstrating potential for a more cost-effective approach.

## 1. Introduction

Sleep posture, which plays a pivotal role in determining sleep quality, has gained considerable attention in the field of sleep medicine. It has been found that a supine posture may increase the risk of obstructive sleep apnea (OSA), while a lateral posture could potentially reduce such risks [1]. Additionally, the *IEEE Sensors Journal* [2] underscores the significance of regular positional adjustments for bedridden patients to prevent ulcers and bedsores, pointing to the necessity of sleep posture monitoring. Despite polysomnography (PSG) being recognized as the definitive standard for assessing sleep posture [3], its limitations due to high costs, time consumption, and the requirement for professional oversight restrict its utility for continuous monitoring. Consequently, there is significant value in developing an accurate and portable method for sleep posture detection.

Home sleep tests (HSTs) provide a convenient and cost-effective way for individuals to monitor their sleep conditions at home. HST devices mainly fall into two primary categories: wearable devices and non-contact devices. Wearable devices, exemplified by chest straps equipped with gyroscopes [4,5] and wristbands integrated with accelerometers [6,7,8], enable the simultaneous detection of sleep posture and monitoring of physiological parameters. However, their reliance on being placed on specific body parts, such as the wrist or chest, may lead to inaccuracies due to loose or incorrect placement, making them impractical for long-term use [9,10]. This limitation has prompted researchers to increasingly focus on non-contact devices, which can monitor sleep without interfering with the user’s normal activities.

To address these limitations, non-contact devices for monitoring sleep have been developed, such as camera-based, radar-based, and pressure-based systems. Camera systems using red-green-blue (RGB) or red-green-blue-depth (RGB-D) imaging technologies provide detailed visuals of sleep patterns but are often restricted by their high computational cost, lighting conditions, and obstructions by clothing or bedding [8,11,12,13]. These cameras also pose potential privacy risks. Radar devices, typically employing continuous wave technology, can track sleep postures by identifying signal reflection changes during movements such as rolling [9,14,15,16,17]. These devices can distinguish supine from lateral positions but frequently encounter difficulties in differentiating finer details, like left versus right lateral positions, and separating signals from one’s pulse and respiration [18].

Pressure-based systems, particularly those employing dense sensor arrays, have been increasingly recognized as a promising method for detecting sleep postures by effectively mapping the body’s pressure distribution. Early research utilized large mattresses equipped with thousands of sensors to monitor pressure changes during sleep [19,20,21,22,23,24,25,26,27]. Sophisticated analysis methods, including support vector machines (SVMs), the k-nearest neighbors (KNN) algorithm, convolutional neural networks (CNNs), and ResNet, have been applied to these pressure maps, achieving accuracy rates exceeding 95%. However, the considerable size and high cost of these sensor arrays have become barriers to developing portable devices. In response, researchers like J. Liu and Y. Chen have developed innovative devices with sparse sensor arrays which focus on measuring body pressure with fewer sensors [18,28,29,30,31]. These systems, with sensors strategically placed in proper areas, manage to achieve nearly 90% accuracy in detecting sleep postures. Despite this, they overly rely on static pressure distribution images. This reduction in sensor resolution leads to a significant loss of critical detail, weakening the model’s ability to extract complex features and limiting its adaptability and generalizability. Furthermore, their performance is evaluated using lab-generated data rather than real sleep scenarios, making them unsuitable for handling complex signal environments. Overall, these technologies are not robust for daily home sleep monitoring across diverse populations.

In this study, we propose S^3^CNN, a novel sparse sensor-based spatiotemporal convolutional neural network tailored for sleep posture detection. The network uses a strategically limited set of piezoelectric ceramic sensors to efficiently capture pressure signals. To compensate for the limited number of sensors, the S^3^CNN utilizes advanced feature extraction techniques to extract critical indicators from multi-channel vibrational data. The network enhances adaptability by combining spatial and temporal data processing. It uses two shallow 2D convolutional neural networks (Conv2Ds) to process spatial patterns within cardiorespiratory activity maps and two shallow 1D convolutional neural networks (Conv1Ds) to handle temporal features from the heart and respiratory rates. This approach not only facilitates robust detection of sleep postures but also improves the model’s generalizability by capturing dynamic physiological changes over time, reducing the dependence on static data. Testing on datasets has demonstrated that the S^3^CNN achieves superior performance with a minimal number of sensors. The primary contributions of this study are outlined as follows:Using piezoelectric ceramic sensor arrays centered in the chest area, our 58 cm × 28 cm mattress with only 32 sensors enables precise sleep posture detection.We developed a model-based feature extraction method which enhances the interpretability of physiological parameters by correlating them with electrical signals, improving the performance of sleep posture classification.The S^3^CNN architecture, with shallow Conv1Ds and Conv2Ds, effectively integrates dynamic temporal and static spatial features, boosting detection accuracy.Validated on a real-world sleep dataset, the S^3^CNN achieved 93.02% accuracy with a minimal sensor array, demonstrating excellent stability in three rounds of 10 fold cross-validation.

## 2. Hardware Construction

We chose piezoelectric ceramic materials for our monitoring mattress due to their flat shape, high sensitivity, low cost, and excellent high-frequency response. As shown in Figure 1a, these sensors consist of a circular piezoelectric element and a brass plate at the base. When subjected to varying forces, they generate a faint current based on the piezoelectric effect. For stable connectivity, each sensor (3 cm in diameter) was mounted on a 3 cm × 5 cm printed circuit board. The flexible piezoelectric ceramic array had a planar layout, featuring an 8 × 4 electrode matrix which formed 32 channels. Each row and column was connected by electrodes, with each electrode covering an area of 15 square centimeters (3 cm × 5 cm). The array’s total area (58 cm × 28 cm) matched an adult’s chest width, allowing clear visualization of the vibration distribution details.

The schematic of the readout circuit in Figure 1c contains an analog front end (including the piezoelectric ceramic union and non-inverting amplifier), low-pass filter, analog-to-digital converter circuit, and wireless communication circuit. When subjected to a vertical periodic pressure F(t) with an amplitude $F_{m}$ and angular frequency ω, a micro-current $I_{i}$ was generated as follows:(1)$$I_{i}≅\frac{dQ_{i}}{dt}=\omega d_{33}F_{m}\cos\omega t$$
where $d_{33}$ represents the piezoelectric constant generated by the effect. Given that $d_{33}$ is typically quite small (less than 600 pc/N), the current $I_{i}$ generated by the piezoelectric effect with an electric field and pressure in the vertical direction is rather weak. Here, $Q_{i}$ is the charge generated by the external force.

To ensure effective detection and processing of this signal, it is necessary to amplify the feeble current through the non-inverting amplifier, as shown in Figure 1b. In this amplifying circuit, the piezoelectric ceramic sensor is equivalent to an alternating current (AC) power source characterized by an equivalent resistance $R_{1}$ and equivalent capacitance $C_{1}$. Here, $C_{2}$ represents the input capacitance of the amplifier, and $R_{2}$ is the input resistance of the circuit. The input voltage $U_{i}$, the output voltage $U_{im}$, and $U_{o}$ can be calculated as follows:(2)$$U_{i}=\overset{•}{I_{i}}\frac{R_{M}}{1+j\omega R_{M}C_{M}}=j\omega d_{33}\overset{•}{F_{m}}\frac{\frac{R_{1}R_{2}}{R_{1}+R_{2}}}{1+j\omega \frac{R_{1}R_{2}}{R_{1}+R_{2}}\left(C_{1}+C_{2}\right)}$$
(3)$$U_{im}=\left|U_{i}\right|=\frac{d_{33}F_{m}\omega R_{M}}{\sqrt{1+{\left(\omega R_{M}C_{M}\right)}^{2}}}$$
(4)$$U_{o}=\beta U_{im}\cos\omega t$$
where $\overset{•}{I_{i}}$ and $\overset{•}{F_{m}}$ are the complex forms of $I_{i}$ and $F_{m}$, respectively, and $R_{M}$ and $C_{M}$ are the equivalent input resistance and input capacitance of the amplifying circuit, respectively. The output voltage $U_{o}$ can be obtained by an operational amplifier with a gain of β. Thus, we can ascertain the relationship between the amplitude of periodic vibrations and the output AC voltage.

## 3. Methods

Figure 2 shows the overall procedure of sleep posture detection in this study, which includes signal processing, feature extraction, and model training. The details are elaborated below.

### 3.1. Signal Processing

The collected raw signal, containing various physiological data, are often subject to motion artifacts. To precisely analyze the cardiopulmonary features within these mixed signals, a signal processing module was employed. This module consisted of two components: signal decomposition and artifact identification. Signal decomposition isolates cardiopulmonary components from the mixed signals, while artifact identification separates disturbed samples from undisturbed samples, ensuring the accurate extraction of reliable cardiopulmonary features [17].

#### 3.1.1. Signal Decomposition

Respiratory and cardiac activities, including breathing patterns and heart pulsations, generate subtle vibrations detected as composite signals by the sensors. For precise assessment of heart and lung activity, it is crucial to isolate these physiological signals from the aggregated data. The pressure signal produced by respiration typically falls within the frequency range of 0.1–0.8 Hz [32,33], whereas the cardiovascular pressure signal, known as the ballistocardiogram (BCG), fell within the frequency range of 0.8–15 Hz [34]. To extract these two types of signals, we employed a Chebyshev type-I finite impulse response (FIR) band-pass filter with an order of 999. The deviation signal was computed by subtracting the respiratory and cardiac signals from the composite signal. Each decomposed sample contained 6000 sampling points, representing one minute of data. Steps 1–3 of Algorithm 1 outline the signal decomposition process, including designing and applying the filters and calculating the energy components.

Figure 3 illustrates the separation of the undisturbed signal. The respiration signal, which was highly periodic, accounted for approximately 90% of the raw signal’s energy. The ballistocardiogram (BCG) signal consists of two components: an enveloping waveform and IJKL waves [35]. The IJKL waves, related to ventricular ejection and aortic flow, occur at a frequency about one-fifth that of the heartbeat [36] and constitute 5–8% of the raw signal’s energy. The deviation signal, marked by its lack of significant periodicity, comprises less than 3% of the energy. Additionally, according to Equations (Equation 2) and (Equation 3), we can determine the relationship between the periodic movements of the heartbeat and respiration and their corresponding signal components.
**Algorithm 1** Pseudocode for signal decomposition and artifact identification **Require:** Time series data (6000 samples) composite_signal1:**Step 1:** Design band-pass filters2:   Respiratory filter FR: 0.1–0.8 Hz, order 9993:   Ballistocardiogram filter FC: 0.8–15 Hz, order 9994:**Step 2:** Apply filters to composite_signal5:   resp_signal = filter(FR, composite_signal)6:   bcg_signal = filter(FC, composite_signal)7:   dev_signal = composite_signal - (resp_signal + bcg_signal)8:**Step 3:** Calculate energy components9:   composite_energy = sum(composite_signal.^2)10:   resp_energy = sum(resp_signal.^2)11:   bcg_energy = sum(bcg_signal.^2)12:   dev_energy =composite_energy - (resp_energy + bcg_energy)13:**Step 4:** Calculate energy entropy14:   resp_ent_energy = energy_entropy(resp_signal, 400, 40)15:   bcg_ent_energy = energy_entropy(bcg_signal, 100, 10)16:   dev_ent_energy = energy_entropy(dev_signal, 10, 1)17:**Step 5:** Calculate approximate entropy18:   resp_ent_approx = calculate_approx_entropy(resp_signal, 2, 0.2 * std(resp_signal))19:   bcg_ent_approx = calculate_approx_entropy(bcg_signal, 2, 0.2 * std(bcg_signal))20:   dev_ent_approx = calculate_approx_entropy(dev_signal, 2, 0.2 * std(dev_signal))21:**Step 6:** Artifact identification based on entropy features22:   features = [resp_ent_energy, bcg_ent_energy, dev_ent_energy, resp_ent_approx, bcg_ent_approx, dev_ent_approx]23:   model = MLPClassifier(hidden_layer_sizes=(12,), activation=’logistic’)24:   artifact_detected = model.predict(features) > 0.525:**Return:** resp_signal, bcg_signal, dev_signal, artifact_detected

#### 3.1.2. Signal Contamination Management

In realistic environments, body movements commonly introduce motion artifacts into the raw signal. PSG and sleep monitoring data indicate that approximately 13% of nighttime sleep involves various body movements. When body movements occur, the sensors in contact with the body will receive irregular, non-periodic, and intensely amplified forces, resulting in a significant amount of motion artifacts which interfere with normal signals [37]. To identify and eliminate samples severely affected by motion artifacts, we classified the samples as “interfered samples” or “uninterfered samples”. As shown in Equation (Equation 5), the piezoelectric sensitivity $S_{v}$, which represents the coefficient relationship between the applied force F(t) and the output voltage $U_{o}$ [38,39,40], allowed us to more intuitively reflect the differences in patterns between the “interfered samples” and “uninterfered samples”:(5)$$S_{v}≅{\left|\frac{\Delta U_{o}}{\Delta F(t)}\right|}_{\Delta t\to 0}\approx \frac{\beta U_{i}}{F_{m}}=\frac{\beta d_{33}\omega R_{M}}{\sqrt{1+{\left(\omega R_{M}C_{M}\right)}^{2}}}$$
where Δt is the unit time (Δt→0); ΔF(t) is the pressure change in the unit time; $\Delta U_{o}$ is the corresponding change in output voltage; and ω is the angular velocity, which varies with the pressure values. The piezoelectric sensitivity can be calculated based on ω:(6)$$S_{v}\approx \left\{\begin{array}{c} \begin{array}{ccc} \frac{\beta d_{33}}{C_{M}} & & \end{array}{\left(\omega R_{M}C_{M}\right)}^{2}\gg 1\\ \begin{array}{c} \frac{\beta d_{33}\omega R_{M}}{\sqrt{1+\omega R_{M}C_{M}}}\begin{array}{cc} & others \end{array} \end{array} \end{array}\right.$$
As demonstrated in Equation (Equation 6), the piezoelectric sensitivity $S_{v}$ remained approximately constant at sufficiently high frequencies ω. For an undisturbed signal, the angular velocities of heartbeats, respiration, and the pulse satisfy the condition ${\left(\omega R_{M}C_{M}\right)}^{2}\gg 1$, making the output voltage linearly proportional to the input pressure force. Conversely, for the disturbed signal, when this condition is not met, $S_{v}$ varies with ω, leading to a nonlinear relationship between the output voltage and the input force.

We employed entropy features, such as energy entropy and approximate entropy (ApEn), to quantify the nonlinearity of the signals, thereby distinguishing between disturbed and undisturbed signals. Typically, respiratory signals have a breathing cycle of about 4 s, BCG signals have a heartbeat cycle of about 1 s, and deviation signals lack periodicity, which external disturbances can disrupt. To capture significant feature differences, we set the energy entropy windows for respiration, the BCG, and deviation to 400, 100, and 10, with steps of 40, 10, and 1, respectively, as shown in Step 4 of Algorithm 1. For the ApEn feature, we used an embedding dimension of 2 and a similarity threshold of 0.2, as outlined in Step 5. Finally, we classified the signals as “interfered” or “uninterfered” using a multilayer perceptron with 12 hidden units. Through our observations of the PSG data, we found that more significant body movements resulted in a greater number of channels with “interfered” signals. To identify the samples affected by body movements, we used the number of “interfered” channels as a threshold. Increasing the threshold from 1 to 7 channels enhanced the performance of motion detection. However, setting the threshold above 7 did not result in further accuracy improvements and significantly decreased the recall rate. Therefore, we empirically set the threshold to 7 for optimal performance.

### 3.2. Feature Extraction

For uninterfered samples, we extracted the respiration and BCG signals using band-pass filters with ranges of 0.1–0.8 Hz and 0.8–15 Hz, respectively, as depicted by the black points in Figure 4. Given that the frequencies of respiration and heartbeats are relatively fixed and fall within the linear response range of our device, we employed the sum of sinusoids function to approximate these real signals. Based on experimentation, a single sinusoidal term was found to be sufficient for fitting the respiration signal. In 200 sets of experiments, the R-squared value of the fitting curve ranged from 0.85 to 0.93, indicating that the fitting function $U_{res}^{∼}$ was highly consistent with the actual respiration signal $U_{res}$. The simulated respiration $U_{res}^{∼}$ is indicated by the red lines in Figure 4a. The corresponding formulas are as follows:(7)$$U_{res}\approx U_{res}^{∼}\approx A_{res}\sin\left(2\pi f_{res}t\right)=\frac{d_{33}\beta }{C_{M}}F_{res}\sin\left(2\pi f_{res}t\right)$$
where $A_{res}$ is the amplitude of the fitting curve and $f_{res}$ is its frequency, also representing the frequency of respiration activity. Meanwhile, $F_{res}$ reflects the intensity of the respiration force.

The black dotted line in Figure 4b displays the BCG signal, which was composed of 600 sampling points extracted from the undisturbed signals. As previously mentioned, the envelope of the BCG signal aligned with the heartbeat cycle, and one envelope cycle consisted of five IJKL waves. Therefore, we approximated the BCG signal using a polynomial with three cosine terms. During the experiment, we found that the coefficients of the first term $A_{bcg}$ and the second term $B_{bcg}$ were quite similar in the fitted polynomials. Additionally, the frequency of the third term $f_{bcg}^{3}$ was approximately equal to the mean frequency of the first and second terms. Therefore, without considering the phase, the simulated BCG signal could be converted into an amplitude modulation (AM) signal. Figure 4b shows the simulated BCG signal obtained by the fitting AM modulation function as a red line. In 200 sets of experiments, the R-squared value of the simulated BCG signal ranged from 0.7 to 0.75, indicating that the simulated BCG signal $U_{bcg}^{∼}$ was consistent with the actual BCG signal $U_{bcg}$. The corresponding formulas are as follows:(8)$$\begin{array}{c} U_{bcg}\approx U_{bcg}^{∼}=A_{bcg}\cos\left(2\pi f_{bcg}^{1}t\right)+B_{bcg}\cos\left(2\pi f_{bcg}^{2}t\right)\\ \begin{array}{ccc} & & +C_{bcg}\cos\left(2\pi f_{bcg}^{3}t\right) \end{array}\\ \approx C_{bcg}\left\{1+\frac{2A_{bcg}}{C_{bcg}}\cos\left[\pi (f_{bcg}^{1}-f_{bcg}^{2})t\right]\right\}\cos\left[\pi (f_{bcg}^{1}+f_{bcg}^{2})t\right] \end{array}$$
(9)$$\begin{array}{c} U_{bcg}\approx U_{bcg}^{∼}\approx U_{Am}\left[1+M_{bcg}\cos\left(2\pi f_{hea}t\right)\right]\cos\left(2\pi f_{bcg}t\right)\\ \approx \frac{\beta d_{33}F_{bcg}}{C_{M}}\left[1+M_{bcg}\cos\left(2\pi f_{hea}\right)\right]\cos\left(2\pi f_{bcg}\right) \end{array}$$
where $U_{Am}$ represents the amplitude of the amplitude-modulated (AM) signal, $F_{bcg}$ is the force from ventricular ejection and vasoconstriction, $M_{bcg}$, the modulation index, indicates the BCG signal’s amplitude relative to the carrier, and the frequencies $f_{hea}$ and $f_{bcg}$ correspond to the heart rate and frequency of IJKL waves, respectively.

We aim to extract the amplitude features $F_{res}$ and $F_{bcg}$ from these equations. However, the presence of the time t and phase-related cosine terms complicates this task. To isolate these features and eliminate the influence of *t* and the phase, we plan to use the Maclaurin series to calculate the instantaneous variables of the signal, as shown in Equations (Equation 10) and (Equation 11):(10)$$\sum_{t=0}^{t=N\Delta t}\Delta \left|U_{res}\right|\approx \frac{2047\beta d_{33}}{1.65C_{M}}F_{res}\left(2\pi f_{res}\right)\left(4f_{res}N\Delta t\right)$$
(11)$$\sum_{t=0}^{t=N\Delta t}\Delta \left|U_{bcg}^{env}\right|\approx \frac{2047\beta d_{33}}{1.65C_{M}}F_{bcg}M_{bcg}\left(2\pi f_{hea}\right)\left(4f_{hea}N\Delta t\right)$$
where the sampling interval Δt=0.01 s of the signals approaches 0, satisfying the Maclaurin series’ expansion condition [41]. Given that NΔt exceeds the cycle durations of both the heartbeat and respiration, this signal can be approximated as an integer multiple of these cycles. $U_{bcg}^{env}$ is the envelope of the $U_{bcg}$ signal obtained using envelope detection.

Equations (Equation 10) and (Equation 11) reflect the approximate relationships between our target features, the respiratory activity intensity $F_{\mathrm{res}}$, and cardiac activity intensity $F_{\mathrm{bcg}}$ with the respiratory signal $U_{\mathrm{res}}$ and the BCG signal $U_{\mathrm{bcg}}$. According to our previous description, β, $d_{33}$, $C_{M}$, and NΔt are constants. The parameters $f_{\mathrm{res}}$ and $f_{\mathrm{hea}}$ are the respiratory frequency and heartbeat frequency obtained during signal processing. $U_{\mathrm{res}}$ is the extracted respiratory signal, and $U_{bcg}^{env}$ is the envelope of the extracted BCG signal. Therefore, we can obtain the respiratory activity intensity $F_{\mathrm{res}}$ and cardiac activity intensity $F_{\mathrm{bcg}}$ from the decomposed respiratory and BCG signals. Our feature extraction method applies the inverse functions of Equations (Equation 9) and (Equation 10), using a model-based approach to approximate signals and extract the target features.

We derived the estimation of the cardiac and respiratory activity intensities for each channel in the uninterfered samples. These intensities, mapped across a 32 channel array, collectively define the spatial characteristics of cardiopulmonary activities, presenting a static view of features. Figure 5 displays these activity intensities across various sleep postures after applying the data augmentation and smoothing techniques. The original features obtained from the 32 channel sensor formed a 4×8 array, which had a resolution too low to effectively extract features using multi-layer CNN convolutions. Moreover, the large spacing between sensors resulted in significant variations in the features, hindering accurate feature extraction. To address these issues, we applied data augmentation by enlarging the original data by a factor of 4 in both dimensions, resulting in a 16×32 image. This process, combined with specialized MATLAB (9.9.0.1467703 (R2020b)) smoothing, enhanced the resolution and reduced variability, facilitating better feature capture by the CNN.

In Figure 5, the intensity map shows the heart and thoracic regions, with darker colors indicating higher activity intensities. In the supine posture (Figure 5a), the sensor array had the most extensive contact area with the thoracic region. In the lateral posture, the contact area between the chest and the sensor array significantly decreased, allowing for a clear distinction between the supine and lateral postures. Although the contour areas of the left and right lateral images are similar, according to the moment balance principle in statics, the tilt direction of the body was typically the opposite of the offset direction of the center of gravity to maintain stability. In the left lateral posture (Figure 5c), the image tilts to the right, but the high-intensity area is on the left side. Conversely, in the right lateral posture (Figure 5b), the image tilts to the left, but the high-intensity area is on the right side. In summary, the cardiac and respiratory intensity features exhibited significant differences under different sleep postures, making them easier to train for accurate classification.

For the temporal features, we separately obtained the amplitude and interval characteristics of the respiration and heartbeat signals as shown in Algorithm 2. For the respiration signal, to avoid misidentifying the peak values, we applied a smoothing window with a size of 5 to the sampled data. Then, we used the zero-crossing method to detect the peaks and record their amplitudes and intervals. To recover and reflect these dynamic changes in a uniform time series, we employed a feature-based interpolation technique. Specifically, we inserted feature data points between adjacent peaks equal to the number of points in their interval to generate a feature sequence consistent with the length of the original signal. This resulted in a 2×6000 matrix of amplitude and interval features for the respiration signal. In contrast to the respiration signal processing, for the BCG signal, we first performed envelope detection to obtain the heartbeat cycle signal rather than directly detecting the peaks. After obtaining the envelope, we followed the same process as that for the respiration signal, using peak detection and feature-based interpolation. This also resulted in a 2×6000 matrix of amplitude and interval features for the heartbeat signal. Thus, we obtained the initial amplitude interval features for both the respiration and heartbeat signals.

### 3.3. S^3^CNN Model

Building on this foundation, we developed the S^3^CNN, a novel network based on the multi-channel convolutional neural network (MCNN) architecture designed to integrate multimodal features effectively. This network processes one-dimensional temporal features, including the inter-beat intervals and respiration peaks. It also integrates two-dimensional spatial features which capture the active distribution of cardiac and respiratory activities across the sensor array. Figure 6 illustrates the architecture of the S³CNN, comprising two primary modules: the MCNN for optimized feature extraction and a classification module for sleep posture detection. The MCNN analyzes data across multiple channels, enhancing feature integration. Subsequently, the posture detection module fuses these integrated features to classify different sleep postures effectively.
**Algorithm 2** Pseudocode for temporal feature extraction and resampling **Require:** Time series data resp_signal, bcg_signal, dev_signal1:**Step 1:** Respiration Signal Processing2:   smoothed_resp_signal = smooth(resp_signal, window_size=5)3:   resp_peaks = zero_crossing_peaks(smoothed_resp_signal)4:   resp_amplitudes = amplitudes(resp_peaks)5:   resp_intervals = intervals(resp_peaks)6:   resp_features = stack_features(resp_amplitudes, resp_intervals)7:**Step 2:** BCG Signal Processing8:   envelope_bcg_signal = envelope(bcg_signal)9:   bcg_peaks = zero_crossing_peaks(envelope_bcg_signal)10:   bcg_amplitudes = amplitudes(bcg_peaks)11:   bcg_intervals = intervals(bcg_peaks)12:   bcg_features = stack_features(bcg_amplitudes, bcg_intervals)13:**Step 3:** Feature-Based Interpolation14:   resp_features = interp_features(resp_features, target_length=6000)15:   bcg_features = interp_features(bcg_features, target_length=6000)16:**Step 4:** Resample Features17:   resp_matrix = resample(resp_features, 90)18:   bcg_matrix = resample(bcg_features, 180)19:**Return:** resp_matrix, bcg_matrix

#### 3.3.1. Multi-Channel CNN Module

As illustrated in Figure 7, the detailed network structure and its components are clearly presented. Table 1 provides a comprehensive overview of the parameters used, including the filter sizes, activation functions, and connections, thereby offering an in-depth explanation of the architecture. The multi-channel CNN effectively processes dynamic temporal and static spatial features in separate streams. The dynamic temporal features captured by the heartbeat and respiration amplitude interbeat were input into the MCNN-Conv1D network as two-dimensional vectors (2 × n), where $M^{1}$ and $M^{2}$ represent the respiratory and cardiac features, respectively, $M^{1}$ combines the amplitude $R_{A}^{1}$ and interval $R_{B}^{1}$ features of respiration, and $M^{2}$ comprises the amplitude $R_{A}^{2}$ and interval $R_{B}^{2}$ features of the heartbeat. On the spatial side, the MCNN-Conv2D network analyzed the distribution of the cardiopulmonary intensity through activity maps. Here, $N^{1}$ and $N^{2}$ encapsulate the cardiac $H_{A}$ and respiratory $H_{B}$ activity maps, respectively. The features processed by the Conv1D and Conv2D networks were carefully aligned to ensure consistent output dimensions. These were then stacked to facilitate a comprehensive assessment of the features.

Based on Algorithm 2, the initial amplitude interval features for the respiration and BCG signals were both 2×6000 matrices. Since the respiratory cycle was approximately 4 s, and the heartbeat cycle was roughly 1 s, such a high sampling frequency was unnecessary. Additionally, this high frequency significantly increased the data volume, placing an excessive burden on the network. To address this, we resampled the data, reducing the respiratory data to 90 points per minute and the cardiac data to 180 points per minute. This resampling yielded respiratory amplitude interval features $M^{1}$ with dimensions of 2×90 and cardiac amplitude interval features $M^{2}$ with dimensions of 2×180. This strategy preserved the dynamic characteristics of each breath and heartbeat while maintaining consistent output sizes across the subnetworks. The inputs $M^{1}$ and $M^{2}$ were then processed by two streamlined neural network architectures, Conv1d-1 and Conv1d-2, which consisted of three layers of one-dimensional convolution followed by global average pooling, as shown in Figure 7. This configuration maintained consistency in feature extraction despite variations in the input dimensions. Through the MCNN-Conv1D mapping function $f_{i}\left(\cdot \right)$, each input $M^{i}$ was transformed into scaled feature vectors $F^{i}$, with the process parameterized by $\theta^{i}$ addressing the diverse physiological data, as shown in Equation (Equation 12):(12)$$F^{i}=f_{i}\left(M^{i},\theta^{i}\right)\begin{array}{ccc} & & i=1,2 \end{array}$$

Unlike MCNN-Conv1D, which processes temporal feature vectors, MCNN-Conv2D specializes in spatial features by analyzing images of the cardiopulmonary intensity. Due to the limited number of sensors, the resolution of the cardiopulmonary activity map was initially only 4×8. To compensate for this low resolution, we enlarged the original images fourfold using bicubic interpolation. Consequently, the inputs for the subnetworks Conv2d-1 and Conv2d-2 within MCNN-Conv2D, denoted as $N^{1}$ and $N^{2}$, respectively, were respiration and heartbeat activity maps scaled to dimensions of 16×32. The architecture of each subnetwork included three two-dimensional convolutional layers, followed by a max pooling layer and an adaptive max pooling layer to ensure standardized output dimensions across different inputs. Through the MCNN-Conv2D mapping function $g_{i}\left(\cdot \right)$, each input $N^{i}$ was transformed into scaled feature maps $E^{i}$ parameterized by $\beta^{i}$, as shown in Equation (Equation 13):(13)$$E^{i}=g_{i}\left(N^{i},\beta^{i}\right)\begin{array}{ccc} & & i=1,2 \end{array}$$

#### 3.3.2. Sleep Posture Detection Using Stacked Features

Fusing multi-modal features is crucial in determining sleep posture. The temporal features $\left[F^{1};F^{2}\right]$ extracted through MCNN-Conv1D and spatial features $\left[E^{1};E^{2}\right]$ extracted through MCNN-Conv2D are both presented as 32×2 matrices. We stacked these features into a 32×4 matrix of stacked features $S=[F^{i};E^{i}]$, which were then flattened along the row dimension to combine these diverse types of information. A sequence of fully connected layers of dimensions 64 and 96, followed by a Softmax function, served as the mapping function h(·), tasked with integrating these features for classification. This process effectively encapsulated the temporal and spatial information within the data. As a result, the sleep posture *G* was identified by applying this mapping function to the stacked features, represented by the following equation:(14)$$G=h(S,\theta^{dp})$$
where $\theta^{dp}$ represents the parameters of the posture detection model, encompassing all relevant layers and functions involved in determining the sleep posture.

## 4. Results

### 4.1. Dataset

This study utilized a dataset approved by the Ethical Committee of Peking Union Medical College Hospital on 20 August 2019 with IRB No. JS-2089. The dataset included 22 participants aged 30–68 years, comprising 15 males and 7 females. Most participants suffered from varying degrees of OSA and diverse body types, including eight overweight and five lean individuals. We collected vibration data near the chest using a sparse sensor array, while PSG was utilized to monitor sleep postures and body movements. Data were collected from each participant over durations ranging from 4.5 to 7 h, totaling 8583 min. Of these, 937 min (10.92%) were affected by body movements (referred to as “interfered samples”), while 7646 min (89.08%) were unaffected (referred to as “uninterfered samples”). Throughout a night, there are typically 15–25 sleep posture changes, with each posture lasting approximately 30 min. Due to individual differences in sleep posture habits, the proportion of each sleep posture varied significantly among the participants, making a small number of individual samples insufficient for robust training. However, on average, the time spent in the supine position was greater than that spent in the right lateral position and slightly greater than that spent in the left lateral position. Overall, the dataset was balanced with respect to sleep postures, including 3806 min (49.78%) of the supine position, 2566 min (33.56%) of the right lateral position, and 1274 min (16.66%) of the left lateral position. It is worth noting that some preliminary findings using this dataset were also reported in our paper, being presented at the Annual International Conference of the IEEE Engineering in Medicine Biology Society (EMBC) [42]. The class distribution of the dataset is presented in Table 2.

To effectively assess the model’s adaptability and generalization, we divided the data from 22 nights into 10 subsets based on the participants for a balanced cross-validation process. Each subset contained data from 2–3 nights, leading to significant differences in the sample distribution. We systematically tested the performance through three rounds of 10 fold cross-validation on these subsets. In each round, the performance was evaluated using the F1 score, accuracy (Acc), recall (Rec), precision (Pre), and AUC value. Each 10 fold cross-validation produced 10 confusion matrices, which were then aggregated to obtain an overall confusion matrix for that round. This approach, known as summing confusion matrices, is commonly used to provide a comprehensive reflection of the model’s performance across all subsets. By summing the confusion matrices from each of these 10 iterations, we obtained a final confusion matrix which accurately represented the model’s overall performance. The total sample size used for these confusion matrices was 7646, which corresponded to the “uninterfered” subset used for training. This method mitigated the variability in sleep posture distribution across different nights and individuals, providing a more robust assessment of the model’s capabilities. To further verify the model’s generalization capabilities, we conducted three rounds of cross-validation and calculated the mean and standard deviation of these metrics, ensuring stability and reliability under various environmental conditions.

### 4.2. Detection Performance

To enhance the performance of the S^3^CNN model, we meticulously initialized the critical hyperparameters, including the learning rate, batch size, epoch count, and kernel initialization techniques. For this study, the initial learning rate was set to 0.001, and the training was limited to a maximum of 100 epochs. We chose a batch size of 48 to achieve a balance between the computational efficiency and processing speed. To combat overfitting, L1 regularization was incorporated into the loss function. Apart from these modifications, other parameters were retained at their default settings within the Keras framework. The details of all hyperparameters used in our model, including their respective values and the rationale for choosing them, are provided in Table 3. Finally, the adaptability and generalization ability of our model were validated across different subsets using three rounds of 10 fold cross-validation.

Figure 8a shows the confusion matrix of the sleep posture detection by the S^3^CNN model. With the uninterfered samples, the accuracy, recall, precision, and F1 score of the proposed method reached 93.02%, 91.96%, 92.65%, and 0.9229, respectively. Additionally, Figure 8b illustrates the ROC curves for different postures, with AUC values for supine, right lateral, and left lateral postures at 0.9418, 0.9288, and 0.9464, respectively. The overall performance, indicated by a total AUC of 0.9382, demonstrates the model’s strong capability in accurately classifying sleep postures.

## 5. Discussion

### 5.1. Ablation Study

In this study, the S^3^CNN model utilizes multi-channel CNN networks to extract spatiotemporal features, effectively combining one-dimensional dynamic and two-dimensional static features to recognize sleep postures. To assess the impact of specific components such as the MCNN-Conv1D and MCNN-Conv2D on the model’s performance and generalizability, detailed ablation experiments were conducted. Importantly, we employed three rounds of 10 fold cross-validation to validate the robustness of different ablation studies, a method which effectively quantifies the stability and variability of our experimental results.

As shown in Table 4 and Figure 9, the MCNN-Conv2D, focusing on spatial features, achieved an accuracy of 87.71±0.69%. This surpassed the MCNN-Conv1D, which utilized temporal features and recorded an accuracy of 77.96±1.37%. Although both individual networks performed poorly in recognizing lateral postures, the ROC curves indicate that the MCNN-Conv2D significantly improved the detection of the left lateral posture compared with the MCNN-Conv1D. The integrated S^3^CNN model, combining the Conv1D and Conv2D networks, outperformed these single-feature networks. It maintained the left lateral detection performance of the MCNN-Conv2D while significantly enhancing the accuracy for the supine and right lateral postures, achieving an accuracy of 92.58±0.44%. In three rounds of 10 fold cross-validation, the process of obtaining the confusion matrix involved summing the confusion matrices and ensuring each resulting confusion matrix included data from all samples. Although the training results for sleep posture prediction using the MCNN-Conv1D, MCNN-Conv2D, and S^3^CNN were not identical each time, the differences were within manageable limits. Specifically, the variance in predictions using the MCNN-Conv1D was within 200 samples, while that for the MCNN-Conv2D was within 100 samples and that for the S3CNN was within 50 samples, indicating that the S3CNN maintained stable performance. Overall, the submodules of the S^3^CNN not only improved the performance of sleep posture detection but also significantly enhanced the model’s adaptability and generalizability.

### 5.2. Performance Comparison

Table 5 compares the S^3^CNN with state-of-the-art sleep posture detection methods. While our method trailed the top performers by a margin of 1–5% in accuracy, it significantly reduced the sensor requirements by one to two orders of magnitude. This reduction substantially decreases the costs related to sensors and computational needs, thereby improving its suitability for integration into embedded devices. Furthermore, deep learning methods for feature extraction [17,23,24,28,29,31,43,44,45] consistently outperform traditional techniques such as the k-nearest neighbors (KNN) algorithm, histogram of oriented gradients (HOG), principal component analysis (PCA), and SVMs [2,19,20,21,22,25,26,27,30,46,47] in terms of performance. This highlights the benefits of using a neural network-based architecture in our approach. However, the excessive depth of deep learning networks will significantly increase the demand for computational resources. Current state-of-the-art methods often rely on numerous sensor nodes and parameters, requiring robust hardware capabilities and limiting widespread deployment. These methods typically use absolute pressure values from the entire body, which are highly susceptible to disturbances. To improve accuracy, they increase the sensor density, raising costs and limiting scalability. In contrast, our S^3^CNN achieved excellent performance with a shallow three-layer network by focusing on features from the chest area. We separated the cardiac and respiratory activity intensities, filtering out irrelevant information and closely associating the features with sleep posture, allowing good performance even at low resolutions. However, relying solely on static image features is insufficient for handling sudden data changes and cannot capture variations over time. Therefore, we also extracted the temporal features from respiratory and cardiac signals and integrated these multimodal features using the S^3^CNN. This integration further enhances the robustness and accuracy of sleep posture recognition, making our method suitable for portable and wearable devices and broadening its potential for real-world applications.

Unlike traditional methods which rely on lab-generated data, our study used datasets from real sleep scenarios, including significant motion artifacts, which accounted for approximately 13% of the data. These motion artifacts notably degrade the accuracy of sleep posture detection, posing a major challenge for real-life applications. We addressed this by employing a model-based feature extraction method to distinguish between interfered and uninterfered samples, significantly mitigating the impact of motion artifacts on the detection accuracy. By integrating static spatial features with dynamic temporal features, our method enhances adaptability and generalizability, achieving stable performance in practical scenarios. To verify the real-world applicability of the S^3^CNN, our device was part of a national project titled “Research on the Safety and Efficacy System and Standard System of Active Health Products and Human Health State Assessment” (2018YFC200148). Over 100 participants tested our device in homes, factories, and nursing homes, collecting more than 500 nights of sleep posture data. User feedback indicated that our device’s monitoring results were consistent with the actual conditions. These data also contributed to the development of standards and the publication of related works on human health state monitoring, demonstrating the practicality and high performance of our method in embedded devices for long-term monitoring. Moreover, we are currently collaborating with industry partners to bring our technology to market.

### 5.3. Analysis of Model Adaptability and Limitations

The proposed SNNN was thoroughly validated using 10 datasets derived from a sleep posture dataset. The validation process involved a 10 fold cross-validation method repeated across three rounds to ensure robustness. Each fold’s training set comprised 7243–7755 samples from 19–20 patients, while the test set consisted of 828–1340 samples from the remaining 2–3 subjects. The S^3^CNN achieved an accuracy rate of 90–93% across the test sets within each fold, demonstrating significant variation in body types, sleep habits, and diverse sleep postures. The consistent performance across all folds and rounds indicates that our model can effectively handle motion artifacts and maintain high performance across diverse long-term datasets. Therefore, it is suitable for widespread use in daily sleep monitoring.

However, the sleep posture dataset provided by the Ethical Committee of Peking Union Medical College Hospital is limited to only 22 subjects. This relatively small dataset imposed significant constraints on our study. A primary limitation is the the infrequency of the prone sleep posture in normal sleep, appearing in less than one percent of cases and making it difficult to collect sufficient prone posture data for training. This scarcity led to the absence of prone posture detection. Moreover, this study filtered out interfered samples without exploring the correlations between sequential sleep postures, resulting in approximately 10 percent of the samples being wasted. Additionally, the lack of exploration into the correlations between sequential sleep postures contributed to the lack of continuity in detecting sleep postures, which adversely affected performance. To address these limitations, future studies will incorporate additional laboratory data to enrich our dataset with prone samples. We also plan to enhance our models by integrating networks which can capture the temporal dynamics across sequences and utilize inter-sample temporal features.

## 6. Conclusions

In this study, we developed an automated sleep posture identification technique using a sparse sensor array of piezoelectric ceramic sensors, employing only 32 sensors. This method effectively detects nuanced pressure disturbances caused by physiological movements, capturing both breath and heart activity. We proposed a synergistic framework named the S^3^CNN, which combines an MCNN-Conv1D to analyze the temporal features and an MCNN-Conv2D to analyze the spatial features from the mixed pressure signals. The S^3^CNN was thoroughly validated across various datasets and achieved performance comparable to advanced methods which utilize a significantly larger array of sensors. This performance not only demonstrates the superior adaptability and generalizability of the S^3^CNN but also offers a cost-effective solution for home sleep monitoring through portable devices. However, our method filters out interfered samples, leaving a portion of the dataset unutilized. Future work will focus on incorporating additional data and enhancing temporal dynamic modeling to further improve accuracy and enable continuous long-term monitoring.

## Figures and Tables

**Figure 1 sensors-24-04833-f001:**
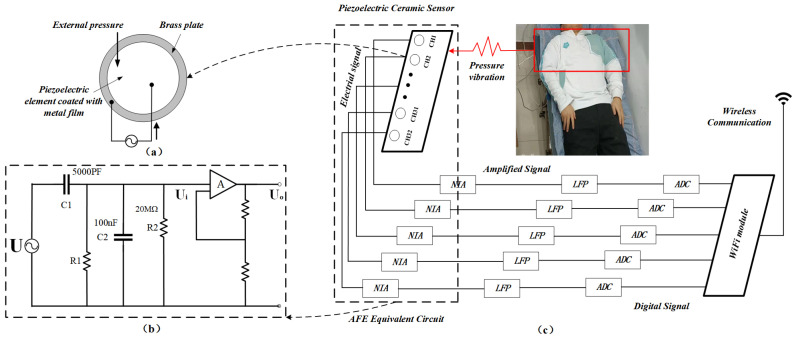
(**a**) Piezoelectric ceramic sensor unit. (**b**) Voltage amplifier circuit. (**c**) Circuit used to scan the pressure distribution.

**Figure 2 sensors-24-04833-f002:**
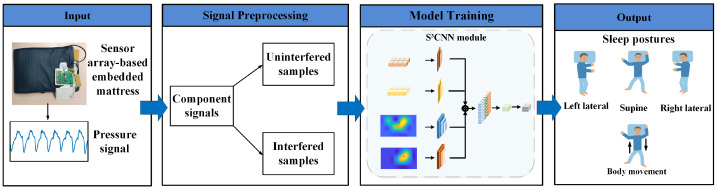
Sleep posture recognition procedure with pressure signal.

**Figure 3 sensors-24-04833-f003:**
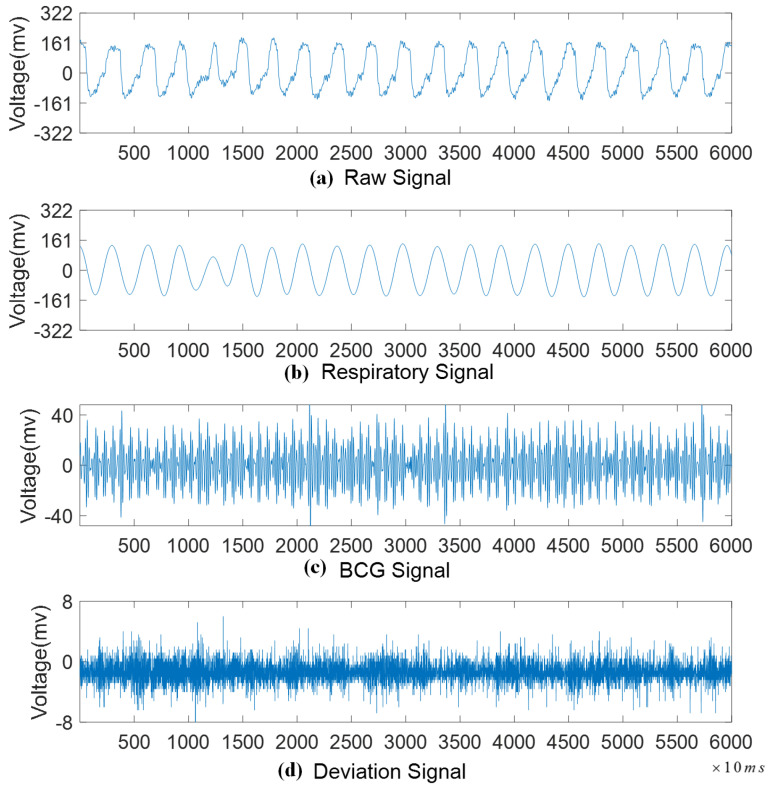
The separation of a one-minute undisturbed signal.

**Figure 4 sensors-24-04833-f004:**
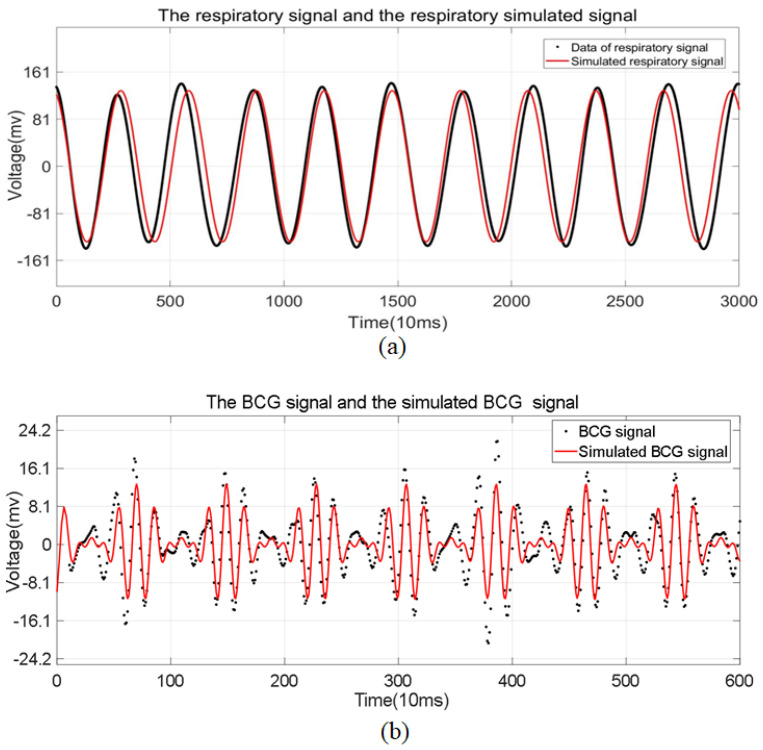
The separation of a one-minute undisturbed signal. (**a**) Resoiratory signal. (**b**) BCG signal.

**Figure 5 sensors-24-04833-f005:**
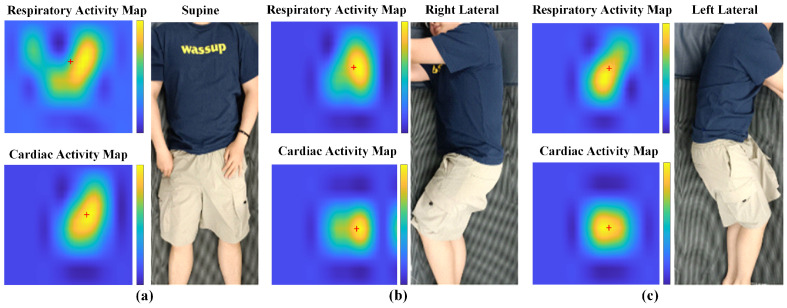
Spatial characteristics of cardiopulmonary activities. (**a**) Supine state. (**b**) Right lateral state. (**c**) Left lateral state.

**Figure 6 sensors-24-04833-f006:**
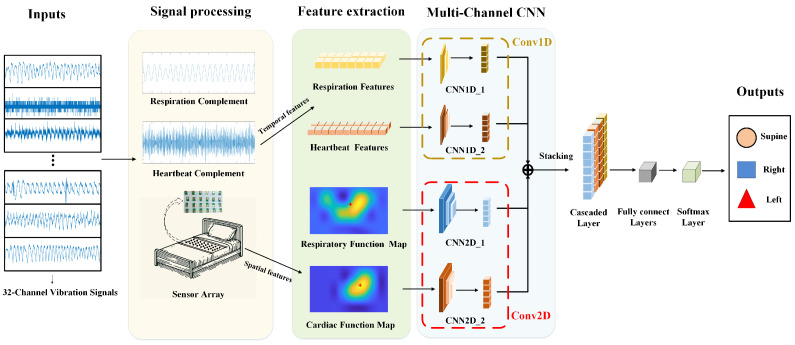
Architecture of S^3^CNN. It mainly includes two modules, namely (1) a multi-channel CNN module and (2) a cascaded layer, with fully connected layers and a Softmax layer.

**Figure 7 sensors-24-04833-f007:**
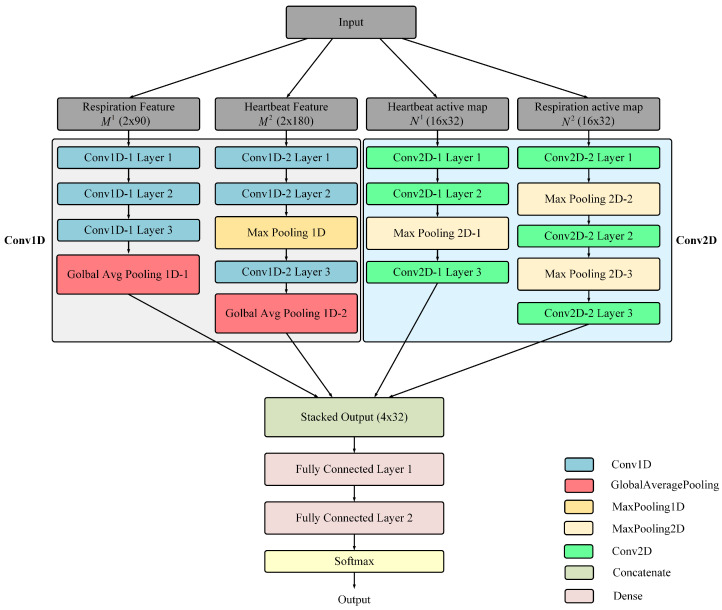
Configuration of S^3^CNN.

**Figure 8 sensors-24-04833-f008:**
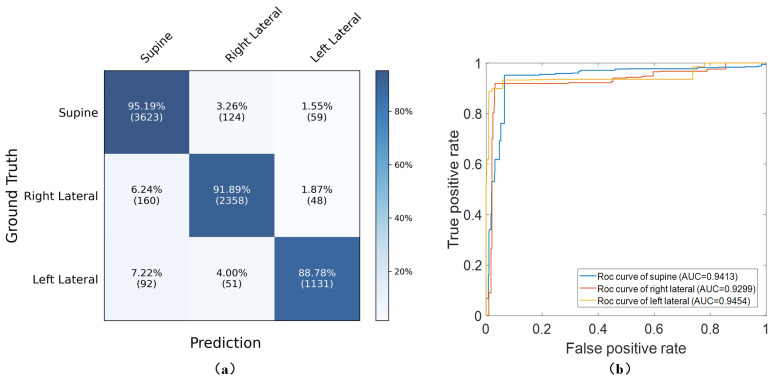
(**a**) Confusion matrix of S^3^CNN. (**b**) AUCs of different sleep postures in S^3^CNN model.

**Figure 9 sensors-24-04833-f009:**
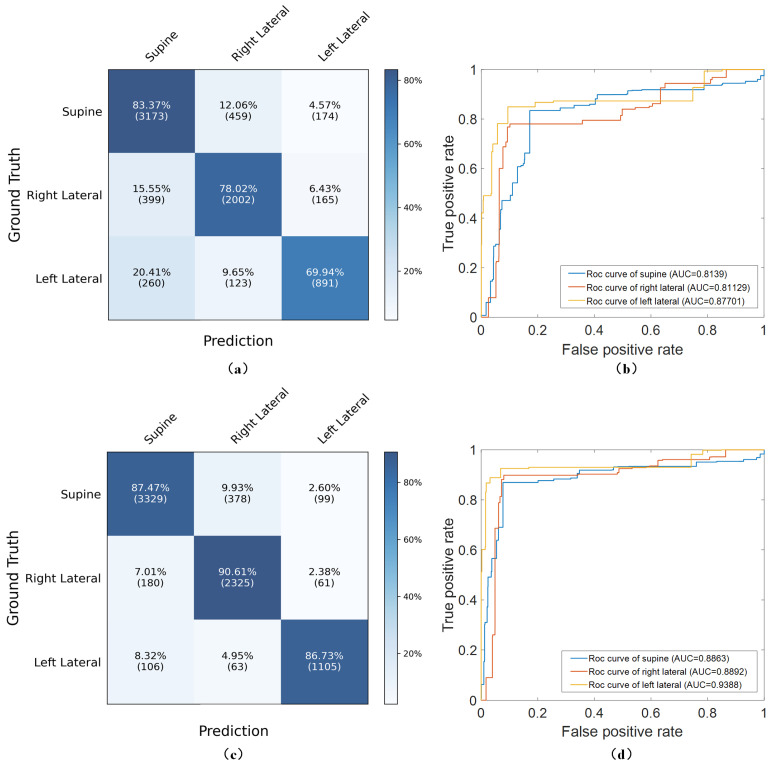
(**a**) Confusion matrix of MCNN-Conv1D. (**b**) AUCs for sleep postures in MCNN-Conv1D. (**c**) Confusion matrix of MCNN-Conv2D. (**d**) AUCs for sleep postures in MCNN-Conv2D.

**Table 1 sensors-24-04833-t001:** Detailed parameter table for S^3^CNN network architecture.

Layer Name	Filters and Kernels	Strides and Padding	Activation
Conv1D-1 Layer 1	filters = 16, kernel = 11	strides = 1, padding = “same”	ReLU
Conv1D-1 Layer 2	filters = 24, kernel = 11	strides = 2, padding = “same”	ReLU
Conv1D-1 Layer 3	filters = 32, kernel = 11	strides = 2, padding = “same”	ReLU
Conv1D-2 Layer 1	filters = 16, kernel = 11	strides = 2, padding = “same”	ReLU
Conv1D-2 Layer 2	filters = 24, kernel = 11	strides = 2, padding = “same”	ReLU
MaxPooling1D	-	pool size = 3	-
Conv1D-2 Layer 3	filters = 32, kernel = 11	strides = 1, padding = “same”	ReLU
Conv2D-1 Layer 1	filters = 6, kernel = (3,3)	strides = (2,2), padding = “same”	ReLU
Conv2D-1 Layer 2	filters = 16, kernel = (3,3)	strides = (2,2), padding = “same”	ReLU
MaxPooling 2D-1	-	pool size = (2,2)	-
Conv2D-1 Layer 3	filters = 32, kernel = (4,2)	strides = (1,1), padding = “valid”	ReLU
Conv2D-2 Layer 1	filters = 12, kernel = (3,3)	strides = (2,2), padding = “same”	ReLU
MaxPooling 2D-2	-	pool size = (2,2)	-
Conv2D-2 Layer 2	filters = 24, kernel = (3,3)	strides = (1,1), padding = “same”	ReLU
MaxPooling 2D-3	-	pool size = (2,2)	-
Conv2D-2 Layer 3	filters = 32, kernel = (4,2)	strides = (1,1), padding = “valid”	ReLU
Fully Connected Layer 1	units = 64	-	relu
Fully Connected Layer 2	units = 96	-	relu

**Table 2 sensors-24-04833-t002:** Description of the dataset.

Sleep Posture	All Samples	Uninterfered Samples	Interfered Samples
Supine	4232	3806	426
Right Lateral	2934	2566	368
Left Lateral	1417	1274	143
Total Samples	8583	7646	937

**Table 3 sensors-24-04833-t003:** Hyperparameter table for S^3^CNN.

Hyperparameter	Value(s)	Rationale
Learning Rate	0.001	Balances convergence speed and training stability
Batch Size	48	Provides a balance between memory usage and gradient estimate stability
Number of Epochs	100	Sufficient for convergence based on initial experimentation
Weight Decay	0.0005	L2 regularization to prevent overfitting by penalizing large weights
Optimizer	Adam	Chosen for its adaptive learning rate and good performance in similar tasks
Dropout Rate	0.5	Helps to prevent overfitting by randomly dropping neurons during training
Activation Functions	ReLU, Softmax	ReLU for hidden layers to introduce non-linearity and Softmax for output layer classification

**Table 4 sensors-24-04833-t004:** Performance comparison in ablation study.

Metric	MCNN-Conv1D for Temporal Features	MCNN-Conv2D for Spatial Features	S^3^CNN for Spatiotemporal Features
Acc	77.96±1.37%	87.71±0.69%	92.58±0.44%
Rec	76.61±1.16%	87.52±0.75%	91.69±0.34%
Pre	76.49±1.08%	86.93±0.68%	92.29±0.36%
F1 score	0.7623±0.0110	0.8734±0.0065	0.9186±0.0043
AUC	0.8145±0.090	0.8912±0.0048	0.9353±0.0029

**Table 5 sensors-24-04833-t005:** Performance comparison with state-of-the-art methods.

Reference	Number of Sensors	Method	Accuracy
Mineharu et al. [19]	64×27=1728	Support Vector Machine	77.1%
Hsia et al. [21]	32×64=2048	Support Vector Machine	83.5%
Qilong et al. [2]	32×32=1024	Support Vector Machine	98.1%
Matar et al. [22]	64×27=1728	HOG + LBP	96.7%
Enokibori et al. [23]	3200	Deep Neural Network	99.7%
Heydarzadeh et al. [24]	32×64=2048	Deep Neural Network	98.1%
Qisong et al. [28]	32×32=1024	CNN-SVM	91.2%
Diao et al. [29]	32×32=1024	RestNet	95.1%
Georgios et al. [20]	32×32=1024	PCA-HMM	90.3%
Xu et al. [25]	64×128=8192	K-nearest Neighborhood	91.2%
Yousef et al. [26]	32×64=2048	K-nearest Neighborhood	97.1%
Zhangjie et al. [30]	32×24=768	HOG-PCA	89.0%
Jen et al. [31]	11×20=220	CNN	96.9%
**Our method**	4×8=32	**S^3^CNN**	**93.0%**

## Data Availability

Enquiries about data availability should be directed to the first authors.

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
