# Peer review of "Smart Sleep Monitoring: Sparse Sensor-Based Spatiotemporal CNN for Sleep Posture Detection"

_sensors, 2024, doi:10.3390/s24154833_

Round 1

Reviewer 1 Report

Comments and Suggestions for Authors

1. Could you provide more details on the specific steps of your signal decomposition and artifact identification process? It would be beneficial to include pseudocode or a flowchart to enhance reproducibility.

2. What specific criteria did you use to classify samples as "Interfered" and "Uninterfered"? Could you elaborate on the threshold values and their justification?

3. The manuscript mentions the initialization of critical hyperparameters. Could you provide a detailed list of all the hyperparameters used in your S3CNN model, along with their respective values and rationale for choosing them?

4. The architecture of S3CNN is briefly described. Could you include a detailed diagram of the network architecture, highlighting the different layers, their parameters (e.g., filter sizes, activation functions), and connections?

5. While the dataset composition is described, additional details would be useful. Can you specify the distribution of sleep postures within the dataset and any data augmentation techniques applied during training?

6. The manuscript reports results from three rounds of 10-fold cross-validation. Could you provide more information on the variance of the performance metrics across these rounds to better understand the model's stability?

7. Your method is compared to various state-of-the-art techniques. Could you provide more insights into why S3CNN, with fewer sensors, achieves comparable performance? Specifically, how does the feature extraction in S3CNN differ from other methods?

8.The manuscript suggests that S3CNN is suitable for portable and wearable devices. Could you discuss any real-world applications or case studies where S3CNN has been implemented or tested outside the laboratory setting?

Reviewer 2 Report

Comments and Suggestions for Authors

In the paper “Smart Sleep Monitoring: Sparse Sensor-Based Spatiotemporal CNN for Sleep Posture Detection” an approach to sleep posture classification is proposed. The paper contains a potentially interest model-based approach to classification but contains the following limitations:

1. Heatmaps in Fugure 5 look like high-resolution heatmaps but based on text it should be 8x4 px images.

2. Next, based on Section 3.2, it is not clear what exactly these features are and how these features are extracted. That is, the section looks like a description of a signal model instead of a feature extractor.

3. The description in Section 3.3.1 is based on Figure 7, and uses notation such as M1, M2, etc., but the figure does not contain such symbols, which makes understanding difficult. Also, these variables are not defined earlier.

4. Probably, due to the previous limitation, the justification of a resampling in lines 237-246 is not clear. 

5. There are some exact results in figures 7 and 8, but in the study a cross-validation approach was utilised without explicit test subset; please, specify a subset used for confusion matrixes. Furthermore, the total sample used for these confusion matrices is 7646 which is equal to the total size of the 'Uninterfered' subset.

6. Next, the features look like interpretable, but there are no attempts to interpret it in the paper.

7. Finally, the proposed network looks like a combination of well-known layers, and novelty does not look enough to segregate the proposed network to a new neural network architecture, especially considering that main novel ideas are in the preprocessing step.

8. Additionally, it makes sense to share a pipeline for the study to improve the reproducibility of the results. 

Therefore, despite the interest idea, the paper suffers from a lack of interpretation and reproducibility, and thus I recommend rejecting the paper.

Round 2

Reviewer 1 Report

Comments and Suggestions for Authors

The authors answered my concerns so well that I have no more questions.

Reviewer 2 Report

Comments and Suggestions for Authors

Thank you for the thorough revision of the paper. The quality of the manuscript has improved significantly. This round if the review is organized as a numerical list where numbers from 1 to 8 is related to Q&A from the first round, besides, I have not noticed new limitations.

1. Thank you for the clarification. Now this process becomes clearer, but it is better to specify exactly which algorithm was used for the image rescaling and for the smoothing.

2. Thank you for the clarification, it is helpful. But it is still unclear what method is used for the signal approximation.

3. Thank you for this valuable improvement.

4. Thank you for this change, but unfortunately, together with p.1, it is still unclear how time series signals became 2x90 or 2x180 feature arrays. It is probably better to represent these features as the left-hand side of an equation or as an algorithm like the one you made for the preprocessing step.

5. Thank you for the clarification.

6. Thank you for the improvement.

7. Thank you for considering my suggestion.

8. Thank you for the changes made.

Therefore, despite the dramatically increased quality of the paper, it is strongly required to fill a gap in methodology description, namely feature extraction and NN input. Thus, the paper requires additional revision.
